# Protocol for the melatools skin self-monitoring trial: a phase II randomised controlled trial of an intervention for primary care patients at higher risk of melanoma

Katie Mills,[1] Jon Emery,[1,2] Rebecca Lantaff,[1] Michael Radford,[3] Merel Pannebakker,[1] Per Hall,[4] Nigel Burrows,[4] Kate Williams,[1] Catherine L Saunders,[1] Peter Murchie,[5] Fiona M Walter[1,2]

For numbered affiliations see end of article.

**Correspondence to**
Dr Katie Mills;
ko298@medschl.cam.ac.uk

## ABSTRACT

**Introduction** Melanoma is the fifth most common cancer in the UK. Incidence rates have quadrupled over the last 30 years and continue to rise, especially among younger people. As routine screening of the general population is not currently recommended in the UK, a focus on secondary prevention through early detection and prompt treatment in individuals at increased risk of melanoma could make an important contribution to improve melanoma outcomes. This paper describes the protocol for a phase II, multisite, randomised controlled trial, in the primary care setting, for patients at increased risk of melanoma. A skin self-monitoring (SSM) smartphone 'App' was used to improve symptom appraisal and encourage help seeking in primary care, thereby promoting early presentation with skin changes suspicious of melanoma.

**Methods and analysis** We aim to recruit 200 participants from general practice waiting rooms in the East of England. Eligible patients are those identified at higher melanoma risk (using a real-time risk assessment tool), without a personal history of melanoma, aged 18 to 75 years. Participants will be invited to a primary care nurse consultation, and randomised to the intervention group (standard written advice on skin cancer detection and sun protection, loading of an SSM 'App' onto the participant's smartphone and instructions on use including self-monitoring reminders) or control group (standard written advice alone). The primary outcomes are consultation rates for changes to a pigmented skin lesion, and the patient interval (time from first noticing a skin change to consultation). Secondary outcomes include patient sun protection behaviours, psychosocial outcomes, and measures of trial feasibility and acceptability.

**Ethics and dissemination** NHS ethical approval has been obtained from Cambridgeshire and Hertfordshire research ethics committee (REC reference 16/EE/0248). The findings from the MelaTools SSM Trial will be disseminated widely through peer-reviewed publications and scientific conferences.

**Trial registration number** ISCTRN16061621.

### Strengths and limitations of this study

► The MelaTools Skin Self-Monitoring (SSM) Trial is among the first, in the UK or internationally, to risk stratify the primary care population in order to target an intervention to improve timeliness of melanoma detection.

► It uses digital technology to encourage people at higher risk of melanoma to self-monitor their skin and present to their physician with any concerns.

► The trial aims to establish the feasibility and acceptability of undertaking SSM among people at higher risk of melanoma; it will also report on consultation rates and includes 6-month and 12-month participant follow-up.

► As this is a feasibility trial, the main limitation is that the small sample size will limit meaningful interpretation of the clinical outcomes; furthermore, the sample may not be fully representative of the UK adult population as it excludes people who do not own a smartphone, and those with physical disorders severe enough to hamper smartphone use.

► Information from this trial will enable planning of a larger phase III trial to further assess the impact of the use of a smartphone App to encourage SSM on clinical outcomes for melanoma.

## INTRODUCTION

Malignant melanoma is the leading cause of skin cancer deaths in the UK with 2459 in 2014.[1] Melanoma skin cancer incidence has quadrupled over the last 30 years, and continues to rise.[2] Although melanoma is more common with increasing age, around half of melanoma in the UK each year are diagnosed in people aged <65.[3 4] Risk factors include fair skin, family history of melanoma, multiple naevi and sun damage. Melanoma is associated with significant morbidity, and the

thickness of the lesion at diagnosis is the most important prognostic factor: stage 1 disease has 5-year survival rates of over 95%, compared with <25% for stage 4 disease.[5] The UK has lower 1-year and 5-year melanoma survival rates than comparable countries in Europe.[6] Diagnostic delays are thought to contribute to this, and there is evidence of avoidable delay.[7] Evidence from the SCREEN project in Germany, conducted between 2003 and 2004, suggested that population screening may have an impact on melanoma incidence and 5-year mortality.[8 9] However, this evidence is controversial, and routine screening of the general population is not currently recommended anywhere worldwide, although some countries (Australia, New Zealand, Germany, the Netherlands and the UK) recommend regular skin checks and/or self-examination for certain subsets of patients at increased risk of melanoma.[10] Therefore, a focus on targeted population screening could make an important contribution to melanoma outcomes through early, timely detection and prompt treatment.

Patient pathways to presentation and management in primary care are key determinants in cancer outcomes. When compared with people diagnosed with other cancers, those diagnosed with melanoma have the second longest median time between first noticing a symptom and presenting to primary care (patient interval).[11 12] The patient interval could potentially be reduced by providing patients with clear information on the signs and symptoms of melanoma and guidance on monitoring skin changes,[13] through community campaigns such as the Be Clear on Cancer skin cancer campaign.[14] More targeted approaches could focus on individuals at higher risk.

Smartphones offer considerable potential to promote earlier presentation by people with cancer, including melanoma.[15] More than 70% of the UK adult population now own a smartphone (Techtracker), and there are many smartphone applications ('Apps') focusing on health issues, presenting new opportunities for risk assessment, symptom appraisal, monitoring symptoms and signs over time, and cues to seek professional advice. Furthermore, up to 75% of melanomas are detected by people or their family/friends, rather than healthcare professionals. To maximise the effectiveness of skin self-monitoring (SSM), the person performing the examination should be able to identify skin changes and features of skin lesions which could indicate melanoma, yet recent studies have demonstrated that suspicious signs of skin cancer may not be widely known.[13 16] Important international evidence has shown that it is possible to educate patients and the public: newly diagnosed Italian melanoma patients who performed self-skin examination were found to have thinner tumours,[17] and Australian patients with melanoma were found to adhere to medical advice on skin self-examination during follow-up care.[18] A recent Scottish study has demonstrated that patients with melanoma are prepared to use digital technology to support them in conducting SSM during follow-up.[19] We therefore set out to explore using mobile technology for SSM

among people at higher risk of melanoma in the primary care setting to encourage timely consultation for possible melanoma.

### Preliminary phase I research
Guided by the Medical Research Council framework on developing and evaluating complex interventions,[20] three phase I studies were conducted to (1) assess the feasibility and acceptability of defining a population at higher risk of melanoma from UK general practice; (2) review the availability of suitable smartphone SSM Apps and (3) pilot the intervention.

### Assessing the feasibility and acceptability of defining a population at higher risk of melanoma from UK general practice
The MelaTools-Q Study findings have already been reported.[21] Briefly, 7742 patients (86% of those approached) were recruited from the waiting rooms of 22 general practices in three UK regions: Eastern England, Northeast Scotland and North Wales. Using tablet computers, they completed an electronic questionnaire incorporating the Williams melanoma risk prediction model,[22] including items relating to seven risk factors: gender, age, natural hair colour at age 15, number of severe sunburns aged 2–18, prior non-melanoma skin cancer, number of raised moles on both arms and density of freckles on both arms before age 20. The study showed that, after weighting to the age and sex distribution, the lower and upper quartile cut-offs used by Williams *et al*, 25 and 34, would allow between 4% and 20% of the population to be identified as higher risk, and those groups would contain 30% and 60%, respectively of individuals who would develop melanoma. Therefore, real-time risk assessment for melanoma in UK primary care is both feasible and acceptable.

### Reviewing the availability of suitable smartphone SSM Apps
Our initial review, conducted in July 2014, identified 39 smartphone Apps available from Apple and Android App online stores.[23] However, when the same search strategy was re-run in September 2015, eight new SSM Apps were identified, while eight of the previously identified SSM Apps were no longer available. Among the 39 newly identified SSM Apps, 11 were not available in both Apple and Android App stores; 10 did not include a monitoring function; 5 did not include photography of skin lesions; 5 had poor functionality; 2 were not available in England; 2 were classified as medical devices; 1 was under development, 1 one had no reminder feature, deemed vital for monitoring functionality. Two SSM Apps were considered potentially most suitable to include in the intervention (see figure 1).

### Piloting the intervention
Qualitative research, using focus groups and interviews, was conducted with individuals at increased risk of melanoma to provide in-depth understanding of consumer views on the usefulness and usability of the two selected SSM Apps. Eight people at increased risk of melanoma

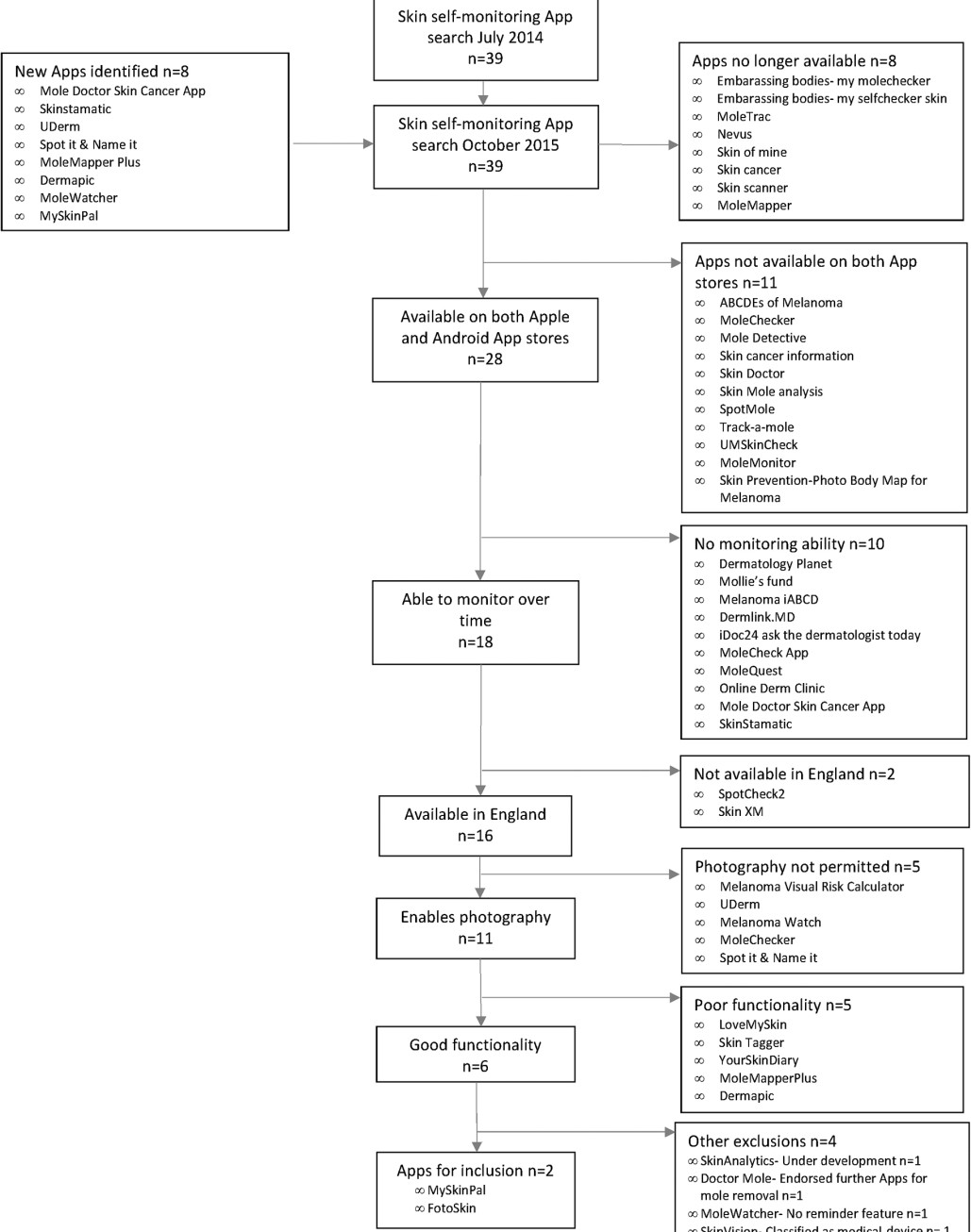

**Figure 1** The availability and suitability of smartphone skin self-monitoring (SSM) Apps for the MelaTools SSM Trial.

were recruited for two focus groups from a local melanoma patient support group; three of their relatives also took part. People were also recruited in the waiting rooms of two suburban general practices and completed the Melatools-Q risk assessment tool.[21] Five identified to be at increased risk of melanoma were invited to participate in two other focus groups. A total of 16 participants tested at least one of the SSM Apps. The participant characteristics are shown in table 1.

The four focus groups were conducted by an experienced qualitative researcher (KM) and research assistant (RL), either at the recruiting general practice or university premises; each lasted approximately 90 min, was audiorecorded and the recordings transcribed

verbatim. Each participant was initially helped to load the two SSM Apps on their smartphones, and given time to familiarise themselves with their use. Thematic analysis[24] showed that, overall, participants in all four groups and at all strata of increased risk of melanoma felt that using a SSM App would be helpful and achievable. Many felt that using a SSM App would enhance their current SSM habits: 'I look at the moles and I just keep an eye on them' *(Male, above-population risk, group 3)*. Many also felt that using a SSM App could help reduce their worry about developing skin cancer: 'cos my dad had cancer recently so [my mother] says we're at a higher risk, so she goes on about that' *(Female, above-population risk, group 4)*. Some felt that using a SSM App would guide their help-seeking

**Table 1** Piloting the intervention–consumer characteristics

|  | Participants (n=16) |
|---|---|
| **Risk of melanoma** | |
| High risk of melanoma (treated melanoma) | 8 |
| At above population risk | 5 |
| Unknown risk (relative of high risk participant) | 3 |
| **Gender** | |
| Male | 8 |
| Female | 8 |
| **Age** | |
| <35 years | 2 |
| 36–44 years | 1 |
| 45–54 years | 3 |
| 55–64 years | 7 |
| 65–74 years | 3 |
| **Ethnicity** | |
| White British | 15 |
| White other | 1 |
| **Education (highest qualification)** | |
| GCSE or equivalent* | 1 |
| A level or equivalent† | 3 |
| Vocational | 2 |
| Undergraduate degree | 3 |
| Postgraduate degree or professional qualification | 7 |
| **Employment status** | |
| Student | 1 |
| Work part time | 5 |
| Work full time | 4 |
| Homemaker | 1 |
| Retired | 5 |
| **Index of Multiple Deprivation quintiles** | |
| Least deprived 1 | 8 |
| 2 | 5 |
| 3 | 2 |
| 4 | 0 |
| Most deprived 5 | 0 |
| Unknown | 1 |

*GCSE, an academic qualification awarded in a specified subject, generally taken in a number of subjects by pupils aged 14–16 in England and Wales.
†A level, same, generally taken aged 16–18.
GCSE, General Certificate of Secondary Education.

behaviour for skin changes or moles: 'It's that knowledge, if it's, you know, three months, whatever it is, if you can actually say look there is a visible change then you're getting a bit of information that they probably could do

with…you've got physical evidence, and then the confidence to go to…' (Male, above-population risk, group 4). Several also commented on additional features on smartphones which might help prevent development of future melanomas: 'I've got a UV index on my phone so I can always check what that is, so putting sun cream on.' (Male, high risk, group 2).

The five participants recruited via primary care were invited to complete a diary 1 and 2 months later, and then participate in a telephone interview after the third month. Diary records and thematic analysis of the interviews (as above) confirmed that participants used many features of both SSM Apps, including taking photographs, mapping these to body parts, comparing photographs over time and using the analysis function (available in only one of the Apps). There was unanimous agreement between participants about the App which was easier to use (MySkinPal), and most reported that they would continue to use it after the study, particularly as they found the App's notifications the most efficient method to prompt them to complete future SSM. Despite this, several also reported that they found taking photographs quite challenging, and that they had to recruit a family member to assist them in taking photographs to compare skin changes over time. Furthermore, while regularly self-monitoring their skin during the study, most participants reported worrying about skin cancer 'sometimes', seeking further information about skin changes or moles via the internet, speaking to friends and family, and visiting their general practice.

In conclusion, this preliminary phase I work enabled consumer choice of the more user-friendly SSM App, resulted in modifications to the language used in the trial consultation and led to the development of the Mela-Tools SSM Trial.

The trial aims to assess the effect of using an SSM App compared with standard information about detecting skin cancer on consultation rates and the patient interval (the time from first noticing a skin change/ mole to consultation with a primary care healthcare professional) among patients at increased risk of melanoma, identified via primary care. In addition, to obtain preliminary estimates of effect across a range of outcome measures to inform selection of the primary outcome for a definitive phase III trial. We hypothesise that primary care patients at higher risk of melanoma will seek help more rapidly after noticing skin changes when using an SSM App compared with using standard written information.

## METHODS AND ANALYSIS
### Study design and setting
The phase II trial is a multisite randomised controlled trial (figure 2), set in 11 general practices across Eastern England. Those who meet the eligibility criteria and consent to participate are randomised 1:1 into either the control or intervention group. Randomisation is being

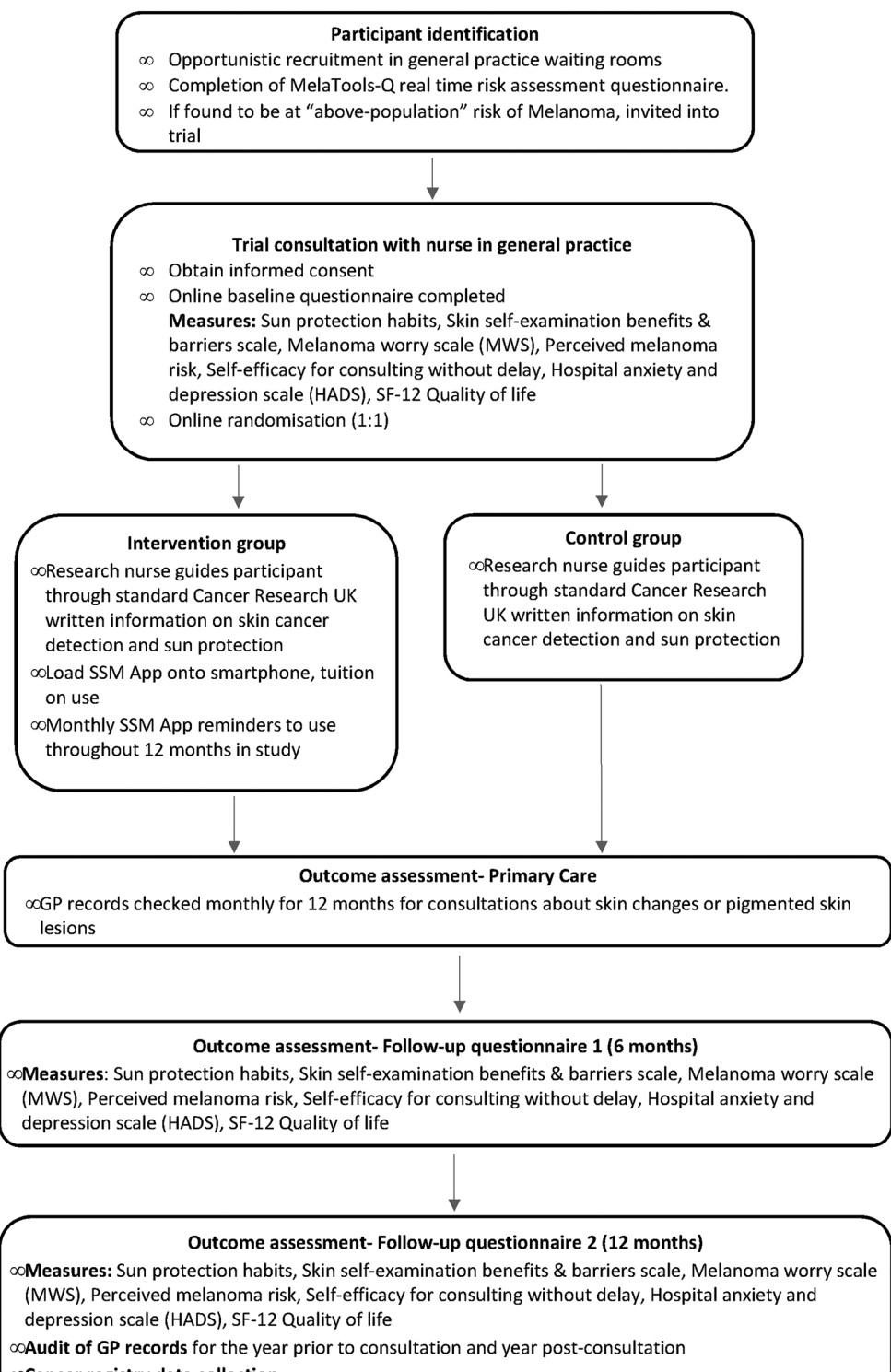

**Figure 2** MelaTools Skin Self-monitoring (SSM) Trial flowchart.

performed using an online system provided by the Clinical Trials Unit based at Kings College London.

### Study participants
#### Eligibility criteria
Eligible participants are individuals aged between 18 and 75 years who own a smartphone (Apple or Android), and, on completion of the Melatools-Q risk assessment tool,[21] are found to be at increased risk of melanoma. Participants are able to read and write English and to give informed consent.

#### Exclusion criteria
Exclusion criteria are previous diagnosis of melanoma, severe psychiatric or cognitive disorders, or a physical disorder severe enough to inhibit the use of a smartphone.

## Participant recruitment

Potentially eligible individuals are approached by research nurses or research assistants in GP waiting areas at different times of the day and different days of the week, in order to ensure that a broad range of ages, gender and educational level are approached. Those meeting initial screening criteria are given a tablet computer that includes a short version of the participant information sheet and an online consent form agreeing to have their risk assessed and be considered for the trial. They complete the MelaTools Q study risk assessment tool[21] which stratifies respondents into 'population risk' (majority) and 'increased risk', using a cut-off score in the Williams model of 25. People at 'population risk' are thanked and given the Cancer Research UK leaflets on risk factors of melanoma and melanoma detection advice. People at increased risk are invited to participate in the trial, and given the full-length trial Participant Information Sheet and an appointment time within 14 days for the primary care nurse trial consultation. Individuals not wishing to take part are asked their reason for declining.

## Control and intervention groups and randomisation

Informed written consent is obtained from all participants at the start of the trial consultation (online supplementary file S1). After completion of a baseline questionnaire, the trial nurse guides the participant through the Cancer Research UK leaflets on 'Be SunSmart cut your cancer risk' and 'Skin cancer—How to spot the signs and symptoms'. Online randomisation is then performed: those randomised to the control group are thanked for their participation and both groups are given the Cancer Research UK leaflets to take home.

## Intervention group

Participants randomised to the intervention group are assisted with loading the SSM App onto their smartphone by the trial nurse, who continues by giving instructions on its use, supported by written instruction sheets. The core functions of the App, MySkinPal, include full body scan, body diagrams, photo source and browser, information on the signs of melanoma, regular reminders, ability to select moles to share with a doctor and motivation to use including awards/achievements. The nurse ensures that the App's monthly reminder notification is switched on to prompt each participant regularly. Finally, each intervention participant is given an Apple Apps store/Google Play voucher (up to £5) to pay for the App.

## Outcomes and measures

The primary outcomes of this trial are *consultation rates* for pigmented skin lesions and the *patient interval*. *Consultation rates* for changes to an existing pigmented skin lesion or concerns about a new pigmented skin lesion for 12 months before the trial and 12 months after the trial consultation will be collected via searching general practice electronic medical records using a clinical notes audit for every participant. The *patient interval* is the time from first noticing a skin change to first consultation. These data will be collected monthly by searches of general practice electronic medical records for all skin changes or pigmented skin lesions presented to their general practitioner (GP) or practice nurse during the 12 months following the trial consultation. When a participant consults about a pigmented skin lesion, they receive a skin questionnaire to complete. This skin questionnaire uses the symptom study instrument modified for melanoma symptoms. The instrument is a participant-completed questionnaire collecting data on symptoms and their duration prior to consultation, validated in prospective studies examining symptom and patient factors associated with longer time to diagnosis for colorectal, lung and pancreatic cancer.[25–27] The skin questionnaire comprises 10 items starting with a free text response to 'What was the first skin change or mole you noticed that made you think something might be wrong?' Each item includes the subitems: 'When did you first notice this?' and 'When did you first tell your GP or nurse?'

Secondary measures and outcomes (measured at baseline, 6 and 12 months) include:

1. *demographics and clinical variables:* age, gender, marital status, postcode, highest education level, occupation, history of skin cancer (melanoma, squamous cell carcinoma, basal cell carcinoma), skin and hair type (density of freckles on arms before age 20, natural hair colour at age 15, number of severe sunburns aged 2–18), number of raised moles on both arms, measured at baseline only,[21 22] collected as part of the baseline eligibility assessment;

2. *sun protection habits scale* developed by Glanz *et al* in the USA for a multicomponent skin cancer prevention programme; this comprises five items measured using a four-point Likert scale, and relating to use of sun protection, sun and sunbed habits, and episodes of sunburn in the previous year,[28] as we hypothesise that the App reminders will prompt people and reinforce the messages on sun protection;

3. *skin self-examination benefits and barriers scale:* validated by Manne and Lessin in the USA among melanoma survivors, and developed from previous work on mammography and family members of patients with colorectal cancer.[29] The benefits scale has seven items ($\alpha=0.71$) and the barriers scale has 10 items ($\alpha=0.74$);

4. *Melanoma Worry Scale (MWS)* validated by Moye *et al* in the USA,[30] and adapted from the Breast Cancer Worry Scale;[31] this measure comprises four items, scored 1 to 4, with possible scores ranging from 4 to 17, and higher scores indicating higher levels of worry

5. *Perceived melanoma risk:* drawn from Manne and Lessin's measures,[29] these two items have been widely used for melanoma and other cancer risk assessments to assess estimated percent risk of developing melanoma, and perceived risk compared with a person of the same age (relative risk);

6. *Self-efficacy for consulting without delay:* A 10-item self-completed scale summed to score 10–100, was

used in a primary care trial for lung symptoms and showed good internal reliability (Cronbach α=0.85).[32 33] It has been adapted for this trial, and reduced to eight items, for example. 'How confident are you that you can make an appointment to see a doctor when…… you can't get an appointment with your usual doctor?';

7. *Hospital Anxiety and Depression Scale:* This 14-item self-completed scale has been widely used to measure distress and has been extensively validated and shown to perform well in a wide range of populations (mean Cronbach α=0.82; sensitivity and specificity 0.80)[34];

8. *12-Item Short Form Survey (SF-12) quality-of-life scale:* a 12-item version of the SF-36 that is widely used and validated to measure functional health and well-being,[35] for example, 'During the past 4 weeks, have you had any of the following problems with your work or other regular daily activities as a result of your physical health?… … '

Other outcome measures include:

9. *trial feasibility and acceptability,* including data on patient recruitment and attrition rates, reasons for attrition, response rates to skin symptom questionnaires for measuring the patient interval to inform decisions about a future phase III trial;

10. *melanoma incidence across participating practices* to contextualise trial findings and after 5 years. This will be identified through GP electronic medical records and the National Cancer Registration Service.

## Measurement timings and study end points

At the trial consultation, all participants will be fully informed about trial follow-up procedures. The participant-reported measures (numbers 1–8 above) are completed in person at baseline, and online at follow-up 1 (6 months) and follow-up 2 (12 months), with the exception of the skin symptom questionnaire as already described, which is sent to participants by postal mail within a month of any consultation for skin changes or pigmented skin lesions.

After trial completion, a final audit of the GP electronic medical records will be run to identify all skin consultations for the 12 months during the trial as well as the previous 12 months. Melanoma incidence findings will also be collected at the end of the 12-month follow-up period and at 5 years. These data will not be collected for participants who formally withdraw from the trial.

## Participant data and study management

All participants are allocated a unique identifying code. Melanoma risk assessment data, baseline questionnaire and follow-up questionnaires data are stored on a custom built database on a secure server hosted by the Outcome Registry Intervention and Operation Network, Clinical Neurosciences Department at the University of Cambridge.

## Sample size consideration

Based on our recent MelaTools Q study with an identical screening step (response rate of 86%), we anticipate that we will need to approach approximately 2000 people from 10 general practice waiting rooms for about 1600 to complete the Melatools Q risk assessment tool.[23] About 25%, 400 people, will be identified as increased risk and be eligible to participate in the trial. Based on previous research, we would expect approximately 50% of these to attend their trial consultation and undergo randomisation in order to reach our target of 200 participants.

We will use information on recruitment, attrition and data completeness, and preliminary estimates of effect sizes where appropriate, to inform the sample size calculation for a future phase III trial.

## Statistical analysis

All randomised patients will be considered eligible for inclusion in the analysis in accordance with the intention-to-treat analysis principle. As this is a phase II trial, fully describing and characterising the extent and nature of the missing data is an important part of the analysis. For the outcome analysis, appropriate methods for dealing with missing endpoint data will be informed by a blinded review of the data; however, assuming that this is appropriate, we plan a series of extreme case-sensitivity analyses, where missing data will be replaced with the 2.5th and 97.5th centiles of the non-missing measured outcomes to assess the maximum potential impact on the results of the trial. The baseline characteristics of the two arms will be described using summary statistics. Possible consent bias will be assessed by comparing demographic and clinical variables of participants with those who declined participation, and possible differential attrition will be assessed by comparing baseline characteristics of those who withdraw or die with those who remain in the trial. In addition, we will analyse the missing data to understand the potential impact on the required sample size, and to assess whether and where possible bias may arise, and therefore be minimised, in the future design for the efficacy phase III trial. We will describe the amount (percentage) of missing data in all outcome measures at baseline and follow-up, and evaluate the relationship between missing data and treatment assignment using unadjusted $X^2$ tests. We will describe whether missing data occur differentially between outcomes. These comparisons will be performed using a two sample t test (or non-parametric equivalent) for continuous variables and $X^2$ test for categorical variables.

The primary analysis will be a comparison between the two groups on the consultation rate for skin changes/pigmented skin lesion using a Poisson regression model. Comparisons between groups on binary secondary endpoints will be performed using logistic regression. Comparisons between groups on continuous secondary endpoints will be undertaken using a linear model that includes the baseline value where applicable. A linear model may not be appropriate for the secondary outcome

patient interval, as this is expected to be right skewed.[36] We will explore the nature and extent of the skewness and analyse patient interval using a linear model if possible, adopting other approaches as appropriate, including transformation or categorisation of the patient interval, or methods for statistical inference based on bootstrap resampling. The analyses performed on the primary and secondary endpoints will be repeated adjusting for additional baseline covariates as part of a sensitivity analysis. Point estimates of the intervention effect will be presented with 95% CIs and two-sided P values. Unadjusted P values from secondary analyses will be interpreted in proper context and be clearly labelled.

## Ethics and dissemination

### Ethical approval

This trial is sponsored by the School of Clinical Medicine at the University of Cambridge. Ethical approval for the MelaTools SSM trial was granted on 11 July 2016 by the Cambridgeshire and Hertfordshire NRES research ethics committee (reference number 16/EE/0248). Health Research Authority approval (HRA) was granted on 22 July 2016. R&D approvals have been obtained from the relevant Trusts. All substantial amendments are approved by the NHS Research Ethics committee responsible for the trial, in addition to approval by HRA and NHS R&D. Investigators are kept up to date with relevant changes via regular management group meetings.

### Data monitoring

Due to the non-medicinal and low-risk nature of the trial, a data monitoring committee will not be needed. The trial steering committee (the CI, collaborators and researchers, statistician and PPIE representatives) will meet six monthly from the start of the trial and will monitor trial progress, approve a data analysis plan, and will ensure the trial runs in accordance with the protocol and applicable standard operating procedures. The CI will take responsibility for data monitoring and ethics, and will be responsible for communicating important protocol modifications to relevant parties.

### Expected findings, implications and dissemination

Twelve-month follow-up will complete in January 2018. We expect that the trial will prove to be both feasible and acceptable to participants without causing them significant anxiety or harm. We do not expect to be able to draw any significant clinical conclusions from this small trial; therefore, we expect to be use these findings to support a funding application for a larger phase III trial aiming to be sufficiently powered to enable evaluation of clinical outcomes. We are also interested in further work exploring the usefulness of different apps for different groups of consumers.

We will submit the findings of our research for publication in highly cited and open access peer-reviewed journals, and the findings will also be presented at national and international conferences. We will also provide a summary of our findings on the MelaTools programme website MelaTools.org.

## Trial progress

The study was registered in the ISRCTN Trials Registry on 17 August 2016, the first participant was recruited on 22 August 2016, enrolment was completed on 6 January 2017 and the participants continue to be followed up in the study, with an estimated finish date of end January 2018.

**Author affiliations**
[1]Department of Public Health and Primary Care, The Primary Care Unit, University of Cambridge, Cambridge, UK
[2]Department of General Practice and the Centre for Cancer Research, Faculty of Medicine, Dentistry and Health Science, Victorian Comprehensive Cancer, Centre University of Melbourne, Carlton, Victoria, Australia
[3]Tennyson House Surgery, Chelmsford, Essex, UK
[4]Cambridge University Hospitals NHS Foundation Trust, Cambridge, UK
[5]Division of Applied Health Science, Centre of Academic Primary Care, University of Aberdeen, Aberdeen, UK

**Acknowledgements** The authors are grateful to the patients and general practices who contributed to the pilot study. The authors would also like to thank the MelaTools Steering Committee for their scientific support and comments throughout the design and set-up of the trial: Pippa Corrie, Katharine Acland, Juliet Usher-Smith, Ed Wilson, Margaret Johnson, Simon Rodwell and Patricia Fairbrother.

**Contributors** FMW and JE developed the initial idea for the trial, and all authors (except MP) have assisted with the development of the protocol, study design and refinement of study materials. PM and JE provided experience of primary care trials, and NB and PH provided clinical expertise. MR led the 2015 SSM Apps review. KM conducted the focus groups and interviews, and led trial preparations with RL and MP, and supported by trials and governance expertise from KW. CS is leading the statistical analysis. KM and FMW wrote the manuscript and revised it in response to critical revisions from all authors. FMW is the guarantor of the study. The International Committee of Medical Journal Editors authorship eligibility guidelines have been followed and no professional writers have been used.

**Funding** This study is supported by the UK Clinical Research Collaboration-registered King's Clinical Trials Unit at King's Health Partners, which is part-funded by the NIHR Biomedical Research Centre for Mental Health at South London and Maudsley NHS Foundation Trust and King's College London and the NIHR Evaluation, Trials and Studies Coordinating Centre. This work was also supported by FMW's Clinician Scientist award (RG 68235) from the National Institute for Health Research (NIHR). The views expressed in this publication are those of the authors and not necessarily those of the National Health Service, the NIHR or the Department of Health. The paper also presents independent research funded/supported by the National Institute for Health Research (NIHR) Collaboration for Leadership in Applied Health Research & Care (CLAHRC) East of England, at Cambridgeshire and Peterborough NHS Foundation Trust.

**Disclaimer** The views expressed are those of the author(s) and not necessarily those of the NHS, the NIHR or the Department of Health.

**Competing interests** None declared.

**Ethics approval** Cambridgeshire and Hertfordshire NHS REC committee, reference 16/EE/0248, protocol version 2 24th June 2016. Requests for the final dataset should be addressedto the corresponding author.

**Provenance and peer review** Not commissioned; externally peer reviewed.

**Data sharing statement** A summary of the results will be disseminated to the study participants. We plan to publish the main trial outcomes in a single paper. Further publications are anticipated after exploring the data in more detail relating to implementation of this novel intervention. Findings will be presented at national and international conferences from late 2018.

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
