## [Reviewer comments · BMJ Open]

ARTICLE DETAILS

TITLE (PROVISIONAL)	Protocol for the MelaTools Skin Self-Monitoring Trial: a phase II randomised controlled trial of an intervention for primary care patients at higher risk of melanoma
AUTHORS	Mills, Katie; Emery, Jon; Lantaff, Rebecca; Radford, Michael; Pannebakker, Merel; Hall, Per; Burrows, Nigel; Williams, Kate; Saunders, Catherine; Murchie, Peter; Walter, Fiona

VERSION 1 – REVIEW

REVIEWER	Anna Finnane The University of Queensland Australia
REVIEW RETURNED	08-Jun-2017

GENERAL COMMENTS	Thank you for the opportunity to review your study protocol. The study is well designed and described and addresses an important need in the field of melanoma early detection. I look forward to seeing the results.
---

REVIEWER	Carolyn Heckman Fox Chase Cancer Center USA
REVIEW RETURNED	15-Jun-2017

GENERAL COMMENTS	Abstract I believe melanoma rates may have started to decline in the US. Please check that this is not also the case in the UK. Strengths and Limitations No limitations are provided. Intro 1. I believe melanoma rates may have started to decline in the US. Please check that this is not also the case in the UK. 2. Suggest briefly noting the controversy around the SCREEN trial and population screening in general (e.g., the USPSTF recommendation). 3. Please briefly provide more info about the Williams melanoma risk prediction model (e.g., number and types of items). Also, was this model validated in the UK or elsewhere? 4. Phase 1, part iii: The text doesn't seem to agree with Table 1 in terms of the number of people at increased risk and the number of groups and strata.
--

Please provide additional clarification about how increased risk, groups, and strata were defined. Also, this section seems like it could be a brief report in itself and could then be referenced more succinctly similar to the other Phase 1 work. Or perhaps some of the details could be put in an appendix or supplementary material.

Methods

1. If “physical disorder severe enough to inhibit self-skin checks” was not included as an exclusion criteria, this should probably be listed as a study limitation.
2. Please provide more info about the number, location, etc. of the GP practices.
3. Expected attrition of 50% between eligibility and consent/baseline 14 days later should probably be listed as a limitation.
4. If material on skin protection isn’t included in MySkinPal, this should be noted. Please provide more info about each of the features of MySkinPal (e.g., how do the full body scan and body diagrams differ from one another?).
5. What search terms or codes will be used for the clinical data? How likely is it that participants would get a skin check with a dermatologist or other practitioner without being noted in the GP records?
6. Please provide more info about the self-efficacy scale (e.g., examples of items). I’m having trouble imagining what the 8 different items would be.
7. Please provide some brief info about the SF-12 (i.e., topics covered by the items).
8. The section on the 10th outcome measure is under-developed.
9. Details about the Symptom Study instrument and related methods are provided in several places. Please move most of the info to the earliest reference or put it all in one place.
10. 200 seems like a small sample size. Power calculations are not provided. What is the expected attrition at 6 and 12 months?
11. Please provide more detail about patient interval data being right skewed.
12. Is it possible to try to find out why there are delays when there are delays to consultation (e.g., are they patient or health system delays) even in a subsample of patients?

After the methods section somewhere, I would like to see a paragraph describing the expected findings and implications. For example, what will be the implications if the consultation rate is high or low, and what would define high or low? Currently, the end of the paper seems short and abrupt.

Clarification for non-expert and/or international readers

1. Write out “MRC”, “PPIE”, “GSCE”, and “IMD” at first use.
2. Define “A Level”.
3. Check phrasing of “tuition on use”.

Figure 2

1. Suggest mentioning the 14 day time period between identification and consultation.
2. I’m confused about the differences between the primary care outcome assessment and the audit of GP records during FU. Please ensure that this is clear in the text as well.

REVIEWER	Elliot Coups, PhD Rutgers Cancer Institute of New Jersey Rutgers, The State University of New Jersey United States
REVIEW RETURNED	26-Jun-2017

GENERAL COMMENTS	 1. Page 2: the first bulleted strength of the study is inaccurate (and claims of “firstness” are rarely warranted). There have been several prior studies that risk stratified primary care populations and targeted individuals at increased risk for melanoma (e.g., PMID 25432953, PMID 23806094, PMID 20167900). 2. Page 3, line 40: please specify the exact percentage of UK adults that own a smartphone. 3. Page 3, line 46: reference 16 is not an appropriate reference in this context. 4. Page 4, line 49: please revise to state that 3 relatives took part and 8 individuals at increased melanoma risk. 5. Page 16, Table 1: “GSCE” should be “GCSE”; add a note to the table to explain the “IMD” acronym and what it represents. 6. Page 6, lines 7-11: please clarify exactly how the randomization schema was devised (e.g., were there any stratification variables? Use of blocking? etc.). 7. Page 7: please clarify whether intervention group participants are also given the leaflets to take home. 8. Page 7, Outcomes and Measures: additional information is needed with regard to the Symptom Study instrument. Additionally, it does not seem appropriate for melanoma incidence across the practices to be considered an “outcome”. 9. There do not appear to be plans to report on the outcomes associated with the consults for pigmented skin lesions (e.g., no action, biopsy, biopsy result, etc.). This seems an important oversight. 10. It is unclear why the 6 and 12 months surveys do not ask intervention group participants to answer questions about their use of, and feedback on, the app. 11. What is the justification for targeting a sample size of 200 in the study? 12. No hypotheses are outlined. They should be added. If there are outcomes for which there are no a priori hypotheses, they should be denoted as such and posited as research questions. No clear rationale is presented for focusing on anxiety, depression, and general quality of life. These are very general outcomes and it is not clear why they would be impacted by the intervention. 13. Are the 6 month and 12 month outcomes considered equal or is one seen as primary? 14. The statistical analysis plan does not state how missing data will be handled. 15. Page 8, line 4: should it be “4 to 16” instead of “4 to 17”?
--

Reviewer: 1, Anna Finnane, The University of Queensland Australia

Comment: Thank you for the opportunity to review your study protocol. The study is well designed and described and addresses an important need in the field of melanoma early detection. I look forward to seeing the results.

Response: Thank you for these kind comments.

Reviewer: 2, Carolyn Heckman, Fox Chase Cancer Center, USA

This protocol describes a small RCT of a skin self-examination “app” based on prior formative work. This study intends to fill an important gap in the literature. More detail and clarification are needed on many points, particularly with regard to the expected findings and implications. However, these items are all fairly easily addressable.

Abstract

I believe melanoma rates may have started to decline in the US. Please check that this is not also the case in the UK.

Response: This is not the case in the UK; in fact, UK rates continue to rise unlike in the US and Australia (Green et al, 2017).

Strengths and Limitations

No limitations are provided.

Response: See above, these have now been included.

Intro

1. I believe melanoma rates may have started to decline in the US. Please check that this is not also the case in the UK.

Response: See above, this is not the case.

2. Suggest briefly noting the controversy around the SCREEN trial and population screening in general (e.g., the USPSTF recommendation).

Response: A further sentence has been added to the Introduction to include this.

3. Please briefly provide more info about the Williams melanoma risk prediction model (e.g., number and types of items). Also, was this model validated in the UK or elsewhere?

Response: More information about the Williams model has been added to Preliminary Phase I research, section (i). The model was originally validated in the US; our recently published paper provides further validation in the UK setting.

4. Phase 1, part iii: The text doesn't seem to agree with Table 1 in terms of the number of people at increased risk and the number of groups and strata. Please provide additional clarification about how increased risk, groups, and strata were defined. Also, this section seems like it could be a brief report in itself and could then be referenced more succinctly similar to the other Phase 1 work. Or perhaps some of the details could be put in an appendix or supplementary material.

Response: The reviewer is correct- the text did not quite match Table 1 so has been corrected. We believe that this section contributes a great deal to the preliminary Phase I research, hence why it is included here and not in a separate publication.

Methods

1. If "physical disorder severe enough to inhibit self-skin checks" was not included as an exclusion criteria, this should probably be listed as a study limitation.

Response: This point has been incorporated into the study limitations.

2. Please provide more info about the number, location, etc. of the GP practices.

Response: We are grateful that the reviewer pointed this out- another sentence describing the practices has been include in the Study design and setting section of the Methods.

3. Exected attrition of 50% between eligibility and consent/baseline 14 days later should probably be listed as a limitation.

Response: We consider this to be reasonable recruitment to a trial requiring a 2-stage screening process for identification of potential participants in the trial, followed by a second attendance for the trial consultation. Therefore, we do not consider this to be a limitation.

4. If material on skin protection isn't included in MySkinPal, this should be noted. Please provide more info about each of the features of MySkinPal (e.g., how do the full body scan and body diagrams differ from one another?).

Response: We very deliberately chose not to include any information about the app, MySkinPal, selected to use in this trial as it is not the app that we are studying. Instead, we are studying the patient behaviour using the app and whether it prompts earlier presentation to their primary care physician. Indeed, we believe that we could have included one of several apps.

5. What search terms or codes will be used for the clinical data? How likely is it that participants would get a skin check with a dermatologist or other practitioner without being noted in the GP records?

Response: In the UK all patients access healthcare via their GP. Therefore, it is very unlikely that participants will get a skin check with another medical practitioner. We have established lists of searches (using Read codes), which are widely used in primary care trials.

6. Please provide more info about the self-efficacy scale (e.g., examples of items). I'm having trouble imagining what the 8 different items would be.

Response: This example has been included in the text: 'How confident are you that you can make an appointment to see a doctor when..... you can't get an appointment with your usual doctor?'

7. Please provide some brief info about the SF-12 (i.e., topics covered by the items).

Response: The SF-12 asks people for their views about their general health, including how well they are able to do their usual activities e.g. 'During the past 4 weeks, have you had any of the following problems with your work or other regular daily activities as a result of your physical health?.....'

8. The section on the 10th outcome measure is under-developed.

Response: We believe that this section succinctly reflects what we aim to do.

9. Details about the Symptom Study instrument and related methods are provided in several places. Please move most of the info to the earliest reference or put it all in one place.

Response: We are not sure what the reviewer means here.

10. 200 seems like a small sample size. Power calculations are not provided. What is the expected attrition at 6 and 12 months?

Response: This is a Phase II feasibility trial, aiming to provide information about feasibility, acceptability and process as well as plausible effect sizes to support the design of a future Phase III trial, including information on attrition and clinical outcomes. 200 patients is sufficient for this purpose. Based on other recent primary care trials of a similar design, we expect less than 10% attrition once participants have been recruited and randomised.

11. Please provide more detail about patient interval data being right skewed.

Response: Prior evidence shows substantial right skew in patient intervals in the UK for many cancers, with median patient interval for melanoma being 21 days, and some patients having intervals of over 234 days. Standard regression approaches to analysing these data may not be appropriate, because this may violate key assumptions particularly including homogeneity of residuals. If this appears to be the case, we will explore alternative approaches to statistical inference.

12. Is it possible to try to find out why there are delays when there are delays to consultation (e.g., are they patient or health system delays) even in a subsample of patients?

Response: This study only examines time to presentation, the Patient Interval, therefore we are not able to comment on patient or health system delays.

After the methods section somewhere, I would like to see a paragraph describing the expected findings and implications. For example, what will be the implications if the consultation rate is high or low, and what would define high or low? Currently, the end of the paper seems short and abrupt.

A short Expected Findings and Implications section has been added.

Clarification for non-expert and/or international readers

1. Write out "MRC", "PPIE", "GSCE", and "IMD" at first use.
2. Define "A Level".
3. Check phrasing of "tuition on use".

Response: These have all been added.

Figure 2

1. Suggest mentioning the 14 day time period between identification and consultation.

Response: This has not been changed as participants were seen between a few minutes and up to 14 days later.

2. I'm confused about the differences between the primary care outcome assessment and the audit of GP records during FU. Please ensure that this is clear in the text as well.

Response: This has been clarified in the text.

Reviewer: 3, Elliot Coups, PhD, Rutgers Cancer Institute of New Jersey, Rutgers, The State University of New Jersey, United States

This manuscript outlines the protocol for a phase II randomized controlled trial of a smartphone app to promote skin self-monitoring among individuals at increased risk for melanoma. Several issues warrant attention.

1. Page 2: the first bulleted strength of the study is inaccurate (and claims of "firstness" are rarely warranted). There have been several prior studies that risk stratified primary care populations and targeted individuals at increased risk for melanoma (e.g., PMID 25432953, PMID 23806094, PMID 20167900).

Response: See above, this has been amended.

2. Page 3, line 40: please specify the exact percentage of UK adults that own a smartphone.

Response: Done

3. Page 3, line 46: reference 16 is not an appropriate reference in this context.

Response: Removed

4. Page 4, line 49: please revise to state that 3 relatives took part and 8 individuals at increased melanoma risk.

and

5. Page 16, Table 1: "GSCE" should be "GCSE"; add a note to the table to explain the "IMD" acronym and what it represents.

Response: See above, done

6. Page 6, lines 7-11: please clarify exactly how the randomization schema was devised (e.g., were there any stratification variables? Use of blocking? etc.).

This text has not been amended as there were no additional features to the randomisation schema, i.e. no stratification variables or use of blocking etc.

7. Page 7: please clarify whether intervention group participants are also given the leaflets to take home.

Response: Yes this has been clarified at the bottom of p6.

8. Page 7, Outcomes and Measures: additional information is needed with regard to the Symptom Study instrument.

Response: We suggest that the Skin Questionnaire could be included as an on-line appendix if the Editor thinks this is suitable for clarification.

8a. Additionally, it does not seem appropriate for melanoma incidence across the practices to be considered an "outcome".

Response: In this small trial it is unlikely that there will be more than a few melanomas diagnosed among 12 participants over 12 months. We consider it appropriate to report the melanoma incidence, and contextualised into regional and national data after 5 years.

9. There do not appear to be plans to report on the outcomes associated with the consults for pigmented skin lesions (e.g., no action, biopsy, biopsy result, etc.). This seems an important oversight.

Response: We agree that pathological findings are important to report too, and have clarified this in the text on clinical outcomes.

10. It is unclear why the 6 and 12 months surveys do not ask intervention group participants to answer questions about their use of, and feedback on, the app.

Response: Our trial is investigating whether giving patients at higher risk of melanoma an app to monitor their moles will prompt more timely presentation with a suspicious lesion. We therefore did not want to possibly contaminate the trial by asking further questions about the app. As before, our interest is primarily in patient behaviour rather than app use.

11. What is the justification for targeting a sample size of 200 in the study?

Response: See response above.

12. No hypotheses are outlined. They should be added. If there are outcomes for which there are no a priori hypotheses, they should be denoted as such and posited as research questions. No clear rationale is presented for focusing on anxiety, depression, and general quality of life. These are very general outcomes and it is not clear why they would be impacted by the intervention.

Response: These are widely used measures in primary care cancer research trials to ensure that an intervention does not cause a participant undue distress. They are of particular value in Phase II trials as we are aiming to establish the acceptability of an intervention as well as the feasibility of a larger trial, powered to identify clinical change. Examples include: Moore et al, Evaluating a computer aid for assessing stomach symptoms (ECASS): study protocol for a randomised controlled trial. *Trials*. 2016;17(1):184, and Walter et al, Effect of adding a diagnostic aid to best practice to manage suspicious pigmented lesions in primary care: randomised controlled trial. *BMJ*, 2012;345:e4110.

13. Are the 6 month and 12 month outcomes considered equal or is one seen as primary?

Response: Equal

14. The statistical analysis plan does not state how missing data will be handled.

Response: The plan explains what steps will be taken

15. Page 8, line 4: should it be "4 to 16" instead of "4 to 17"?

Response: Thank you- this has been amended

Response: We have submitted copies of our revised manuscript in tracked change format. We trust that we have responded adequately to these comments and look forward to hearing from you.

VERSION 2 – REVIEW

REVIEWER	Carolyn Heckman, PhD Fox Chase Cancer Center, USA
REVIEW RETURNED	21-Aug-2017

GENERAL COMMENTS	The authors were moderately responsive to the reviews.
--

REVIEWER	Elliot Coups Rutgers Cancer Institute of New Jersey Rutgers, The State University of New Jersey United States
REVIEW RETURNED	05-Sep-2017

GENERAL COMMENTS	1. Reviewer 2 requested for more information to be included in the text about the app. This would seem important to do.2. As previously requested, please add a sentence or two to describe the Symptom Study instrument.3. The sample size is still not sufficiently justified. This is described as a phase II feasibility trial but the primary outcomes do not related to feasibility. How was a sample size of 200 selected?4. There are still no hypotheses or research questions.5. With regard to missing data, it would be helpful to outline upfront what potential approaches might be suitable for handling missing endpoint data.
--

VERSION 2 – AUTHOR RESPONSE

Reviewer: 3, Elliot Coups, PhD, Rutgers Cancer Institute of New Jersey, Rutgers, The State University of New Jersey, United States

1. Reviewer 2 requested for more information to be included in the text about the app. This would seem important to do.

Response: As we responded last time, we very deliberately chose not to include any information about the app, MySkinPal, selected to use in this trial as it is not the app that we are studying. Instead, we are studying the patient behaviour using the app and whether it prompts earlier presentation to their primary care physician. Indeed, we believe that we could have included one of several apps. The research team continue to believe that this is an important point to make.

2. As previously requested, please add a sentence or two to describe the Symptom Study instrument.

Response: The following sentences have been added to the Outcomes and Measures section:

This skin questionnaire uses the Symptom Study instrument modified for melanoma symptoms. The instrument is a participant-completed questionnaire collecting data on symptoms and their duration prior to consultation, validated in prospective studies examining symptom and patient factors associated with longer time to diagnosis for colorectal, lung and pancreatic cancer (24-27). The skin questionnaire comprises 10 items starting with a free text response to 'What was the first skin change or mole you noticed that made you think something might be wrong?' Each item includes the sub-items: 'When did you first notice this?' and 'When did you first tell your GP or nurse?'

3. The sample size is still not sufficiently justified. This is described as a phase II feasibility trial but the primary outcomes do not related to feasibility. How was a sample size of 200 selected?

Response: One of the primary purposes of this analysis is to estimate parameters to inform the sample size calculation for a future phase III trial for this intervention where appropriate. For example, sample sizes between 24 and 50 have been recommended to estimate standard deviations in feasibility studies to inform future sample size calculations (1,2).

In addition we will use information on preliminary estimates of effect sizes from the primary outcomes for this trial to inform these future calculations. This trial is not powered to detect a difference between the two arms. However the sample size of 100 per arm will allow, for example, a rate ratio of 2 for primary care skin-lesion consultation rate in the intervention compared with the control group to be estimated with 95%CI of 1.6-2.6, 1.4-2.9 and 0.9-4.8 where the baseline consultation rates in the control group are 1, 0.5 and 0.1 per personyear respectively.

4. There are still no hypotheses or research questions.

Response: A hypothesis has been added to the end of the Introduction section.

5. With regard to missing data, it would be helpful to outline upfront what potential approaches might be suitable for handling missing endpoint data.

Response: Our analyses from this feasibility trial will be used to understand the potential impact of missing data on the required sample size, and to assess whether and where possible bias from missing data may arise, and therefore be minimised, in the future design for the efficacy phase III trial.

In addition for the outcome analysis we propose a series of sensitivity analyses to explore the maximum potential impact of missing data on the findings from this trial.

We have added further explanatory text to the Statistical Analysis section:

All randomised patients will be considered eligible for inclusion in the analysis in accordance with the intention-to-treat analysis principle. As this is a phase II trial, fully describing and characterising the extent and nature of the missing data is an important part of the analysis. For the outcome analysis, appropriate methods for dealing with missing endpoint data will be informed by a blinded review of the data; however, assuming that this is appropriate, we plan a series of extreme case-sensitivity analyses, where missing data will be replaced with the 2.5th and 97.5th centiles of the non-missing measured outcomes to assess the maximum potential impact on the results of the trial. The baseline characteristics of the two arms will be described using summary statistics. Possible consent bias will be assessed by comparing demographic and clinical variables of participants against those who declined participation, and possible differential attrition will be assessed by comparing baseline characteristics of those who withdraw or die against those who remain in the trial. In addition we will analyse the missing data to understand the potential impact on the required sample size, and to assess whether and where possible bias may arise, and therefore be minimised, in the future design for the efficacy phase III trial. We will describe the amount (percentage) of missing data in all outcome measures at baseline and follow up, and evaluate the relationship between missing data and treatment assignment using unadjusted chi-squared tests. We will describe whether missing data occurs differentially between outcomes. These comparisons will be performed using a two sample t-test (or non-parametric equivalent) for continuous variables and chi-square test for categorical variables.

We have submitted a copy of our revised manuscript in tracked change format. We trust that we have responded adequately and that this will enable you to progress to publication.

Yours sincerely

Dr Fiona Walter, on behalf of the MelaTools team.

References

1. Sim J, Lewis M. The size of a pilot study for a clinical trial should be calculated in relation to considerations of precision and efficiency. *J Clin Epidemiol* 2012;65:301-308 6.
2. Julious SA. Sample size of 12 per group rule of thumb for a pilot study. *Pharm Stat* 2005;4:287-291.